# The Role of MRE11 in the IL-6/STAT3 Pathway of Lung Cancer Cells

**Ching-Yuan Wu [1,2,3,*], Li-Hsin Shu [1], Hung-Te Liu [1], Yu-Ching Cheng [1], Yu-Huei Wu [4] and Yu-Heng Wu [5]**

1   Department of Chinese Medicine, Chiayi Chang Gung Memorial Hospital, Chiayi 613, Taiwan
2   School of Chinese Medicine, College of Medicine, Chang Gung University, Taoyuan 333, Taiwan
3   Research Center for Chinese Herbal Medicine, College of Human Ecology, Chang Gung University of Science and Technology, Taoyuan 333, Taiwan
4   Department of Biomedical Sciences, Chang Gung University, Taoyuan 333, Taiwan
5   Department of Electrical Engineering, National Sun Yat-Sen University, Kaohsiung 804, Taiwan
*   Correspondence: smbepigwu77@gmail.com

**Abstract:** MRE11 is a pivotal protein for ATM activation during double-strand DNA break. ATM kinase activations may act as lung cancer biomarkers. The IL-6/STAT3 pathway plays an important role in tumor metastasis, including lung cancer. However, the mechanism between MRE11 and the IL-6/STAT3 pathway is still unclear. In this study, we discovered that MRE11 can interact with STAT3 under IL-6 treatment and regulate STAT3 Tyr705 phosphorylation. After the knockdown of MRE11 in lung cancer cells, we discovered that IL-6 or the conditional medium of THP-1 cells can induce the mRNA expression of STAT3 downstream genes, including *CCL2*, in the control cells, but not in MRE11-knockdown lung cancer cells. Moreover, *CCL2* secretion was lower in MRE11-knockdown lung cancer cells than in control cells after treatment with the conditional medium of RAW264.7 cells. In addition, MRE11 deficiency in lung cancer cells decreases their ability to recruit RAW 264.7 cells. Furthermore, MRE11 is a potential target for lung cancer therapy.

**Keywords:** MRE11; lung cancer; STAT3; IL-6; macrophage





## 1. Introduction

Genome integrity maintenance is essential for the appropriate functioning and survival of all organisms. It is also crucial for reasons such as the constant attack on DNA by genotoxic agents and nucleotide misincorporation during DNA replication. In humans, these types of damage induce the development of several inherited disorders, aging, and even oncogenesis. The MRE11 (Meiotic recombination 11 homolog 1)–RAD50–NBS1 (Nijmegen breakage syndrome protein 1) (MRN) is pivotal for ataxia–telangiectasia mutated (ATM) activation during DSB (double-strand DNA break) [1,2]. A previous study revealed that ATM and ATM and RAD3-related (ATR) kinase activations are intact in lung cancer cell lines and may act as lung cancer biomarkers [3]. Moreover, studies have shown that the expression level of the MRN complex is higher in radioresistant small-cell lung cancer cells than in radiosensitive cells [4]. Identifying the critical regulator controlling MRN complex integrity may provide a crucial therapeutic target for lung cancer.

IL-6, a type I interferon (IFN)-activated cytokine, is secreted by T cells and macrophages [5]. Among the cytokines linked to inflammation-associated cancer, IL-6 drives many cancer hallmarks through the downstream activation of the STAT3 (Signal transducer and activator of transcription 3) signaling pathway [6]. STAT3, a critical protein in tumor metastasis, including in the metastasis of lung cancer [7,8], is a valuable biomarker for prognosis prediction and is a therapeutic target in human solid tumors [9]. While cytokine receptors are activated, STAT3 is phosphorylated at Tyr705, leading to the nuclear translocation of activated STAT3 and the induction of downstream target genes such as *CCL2* that promote various cellular processes for cancer progression [10]. Persistently activated or tyrosine-phosphorylated STAT3 (pSTAT3)

is found in 50% of lung adenocarcinomas [11]. Moreover, IL-6 drives many cancer hallmarks through the downstream activation of the STAT3 signaling pathway [6]. Professor Christine Watson reported that inhibitor of DNA binding 1 (ID1, a dominant negative helix–loop–helix protein, transcript variant 1), phospholipid scramblase 1 (PLSCR1), and X-box binding protein 1 (XBP1) are vital downstream genes of STAT3 [12]. Chemokine (C-C motif) ligand 2 (*CCL2*), formerly known as monocyte chemoattractant protein-1, can attract monocytes in vitro [13,14]. In lung cancer, the *CCL2* signaling pathway is the key mechanism through which macrophages activate the growth and metastasis of lung cancer cells by triggering the bidirectional cross-talk between macrophages and these cells [15]. Therefore, blocking the STAT3/*CCL2* signaling pathway may aid in the cessation of lung cancer progression.

Macrophages can promote lung tumor invasion and metastasis [16,17]. Direct mixed co-culture of macrophages and cancer cells could result in stronger transcription factor-mediated activation of signaling [18,19]. Lactate dehydrogenase A (LDHA), regulated by STAT3, is crucial for the invasion and metastasis of malignancies [20,21]. VEGF (vascular endothelial growth factor) is an important signaling protein for vasculogenesis. Previous studies have shown that VEGF is regulated by the STAT3 pathway in several types of cancer, including lung cancer [22,23]. Runt-related transcription factor 2 (RUNX2) has a key role associated with the pathogenesis of some cancers and a downstream gene of the STAT3 pathway [24–26]. Macrophages can migrate and accumulate in distinct tumor microenvironments depending on the chemokine expression pattern [17,27–29]. The *CCL2* signaling pathway is pivotal for the bidirectional cross-talk between macrophages and lung cancer cells during the growth and metastasis of cancer cells [15]. Moreover, *CCL2* is a STAT3 downstream gene [30].

However, the mechanism of interaction between MRE11 and IL6/STAT3/*CCL2* is completely unclear. In this study, we explored the novel role of MRE11 in the IL6/STAT3/*CCL2* signaling pathway in lung cancer. More importantly, we believe that MRE11 might be a novel target for lung cancer diagnosis and treatment.

## 2. Materials and Methods

### 2.1. Cell Culture and Treatment

A549 cells (human lung adenocarcinoma cell line), THP-1 cells (human acute monocytic leukemia cell line), RAW 264.7 cells (mouse macrophage cell line), and 293T cells were obtained from the Bioresource Collection and Research Center, Taiwan. The A549 cells and THP-1 cells were cultured in RPMI-1640 medium (Invitrogen Corp., Carlsbad, CA, USA), supplemented with 10% fetal bovine serum at 37 °C and 5% $CO_2$. The 293T cells and RAW 264.7 cells were cultured in Dulbecco's modified Eagle's medium (DMEM) (Invitrogen Corp., Carlsbad, CA, USA), supplemented with 10% fetal bovine serum at 37 °C and 5% $CO_2$. The indicated cells were cultured to 60–70% confluence before treatment. Then, the medium was then replaced with a fresh medium containing the indicated compounds at the indicated concentrations. The cells treated with water alone were used as untreated vehicle controls.

### 2.2. shRNA Knockdown Assay

For lentiviral short hairpin RNA (shRNA) infection, 293T cells were co-transfected with MRE11 or GFP control shRNA with packing plasmid (deltaVPR8.9) and envelope plasmid (VSV-G) using Lipofectamine 2000 reagent. MRE11-lentiviral shRNA (5′-TTAAAGAACGTC ATCTCGAGATGACGTTCTTTAAGAAATC-3′), and control shRNA (5′-GCAAGCTGACCC TGAAGTTC-3′) were transfected with packing plasmids into 293T cells for 2 days, and virus particles containing MRE11 or control shRNA were used to infect A549 cells. All the infected cells were cultured in a medium containing 2 μg/mL puromycin for 4 days.

### 2.3. Cytokine Membrane Array

The secretion medium of A549 shLeu or A549 shMRE11 cell lines was cultured and collected after being cultured for 24 h. The secretion profiles of cytokines were measured using the Human Cytokine Panel A Array kit (R&D Systems, Minneapolis, Minnesota, United States), according to the manufacturer's instructions. Cell culture supernatants were mixed with a cocktail of biotinylated detection antibodies. Nitrocellulose membranes (spotted with different cytokine antibodies) were then incubated in the sample/antibody mixture. After several washes, streptavidin-HRP and chemiluminescent detection reagents were added, which produced light at each spot proportional to the amount of cytokine bound.

### 2.4. Macrophage Recruitment Assay

Macrophage recruitment analyses were performed as described previously [31]. The human lung cancer cells (A549 shLeu or A549 shMRE11 cells) were cultured for 24 h. The conditioned medium was collected and plated into the lower chamber of transwell plates with a 5 μm pore polycarbonate membrane insert. RAW 264.7 cells ($1 \times 10^4$ cells) were plated onto the upper chamber for macrophage migration assay. After incubation for 24 h, the cells migrated into the bottom are fixed and stained using 1% toluidine blue, and the numbers of migratory cells averaged after counting six randomly selected fields. Each sample was assayed in triplicate. Each experiment was repeated at least twice.

### 2.5. Enzyme-Linked Immunosorbent Assay (ELISA)

The ELISA was performed as described previously [32]. The RAW 264.7 cells were cultured for 24 h. Then, the conditioned medium was collected and plated into a monoculture of lung cancer cells (A549 shLeu or A549 shMRE11 cells). After 24 h, human *CCL2*, IL-6, or IL-8 in the medium were detected by human *CCL2*, IL-6, or IL-8 ELISA kits (eBioscience, San Diego, California, United States) according to the manufacturer's instructions.

### 2.6. Quantitative Real-Time PCR

Total RNA was extracted from lung cancer cells using the TRIzol reagent (Invitrogen, Waltham, Massachusetts, United States, Cat. No. 15596-026) according to the manufacturer's instructions. Reverse transcription was performed using the Superscript first strand synthesis kit (Invitrogen, Waltham, Massachusetts, United States, Number: 11904018). Quantitative real-time PCR analyses using the comparative CT method were performed on an ABI PRISM 7700 sequence detector system using the SYBR Green PCR Master Mix kit (Perkin Elmer, Applied Biosystems, Wellesley, MA, USA) according to the manufacturer's instructions. Following initial incubation at 50 °C for 2 min and 10 min at 95 °C, amplification was performed for 40 cycles at 95 °C for 20 s, 65 °C for 20 s, and 72 °C for 30 s. The primers used were: *CCL2* forward, 5′-GTC TCT GCC GCC CTT CTG TG-3′ and *CCL2* reverse, 5′-GAC ACT TGC TGC TGG TGA TTC TTC-3′. *XBP1* forward, 5′-CCG CAG CAC TCA GAC TAC-3′ and *XBP1* reverse, 5′-TCA ATA CCG CCA GAA TCCAT-3′. *ID1* forward, 5′-CAT TCC ACG TTC TTA ACT GTT CCA-3′ and *ID1* reverse, 5′-ATT CTT GGC GAC TGG CTG AA-3′. *PLSCR1* forward, 5′-ATG ATT GGT GCC TGT TTC CT-3′ and *PLSCR1* reverse, 5′-TCC ACT ACC ACA CTC CTG ATT TT-3′. *LDHA* forward, 5′-ACC CAG TTT CCA CCA TGA TT-3′ and *LDHA* reverse, 5′-CCC AAA ATG CAA GGA ACA CT-3′. *RUNX2* forward, 5′-AGG TAC CAG ATG GGA CTG TG-3′ and *RUNX2* reverse, 5′-TCG TTG AAC CTT GCT ACT TGG-3′. *VEGF* forward, 5′-ACC TCC ACC ATG CCA AGT G-3′ and *VEGF* reverse, 5′-TCT CGA TTG GAT GGC AGT AG-3′. *GAPDH* forward, 5′-TGC ACC ACC AAC TGC TTAGC-3′ and *GAPDH* reverse, and 5′-GGC ATG GAC TGT GGT CATGA-3′. *GAPDH* was used as the housekeeping gene for data normalization.

### 2.7. Western Blot Analysis

For Western blotting, cellular extracts of the human lung cancer cell line (A549 shLeu or A549 shMRE11 cells) treated with indicated compounds for 24 h were prepared according to the manufacturer's instructions. Equal amounts of protein were fractionated on a 6% or 12% SDS-PAGE and transferred to polyvinylidene difluoride (PVDF) membranes. The membranes were then blocked with 5% nonfat dried milk for 30 min and incubated in a primary antibody for 3 h at room temperature. The primary antibodies used were anti-phospho (tyr705)-STAT3 antibody (*p*-STAT3) (cell signaling, ratio: 1:1000), anti-STAT3 antibody (cell signaling, ratio: 1:1000), anti-MRE11 antibody (IP: 1:100; IB: 1:1000, Calbiochem) or anti-β-actin antibody (Santa Cruz, Dallas, Texas, United States, IB: 1:10,000). The primary antibody and secondary antibody were diluted with 1% nonfat dried milk in 1X TBST (Tris-Buffered Saline Tween-20). The blots were washed by 1X TBST and incubated in horseradish peroxidase-conjugated secondary anti-mouse or anti-rabbit antibodies (Santa Cruz, Dallas, Texas, United States, ratio: 1:5000) for one hour at room temperature. After washing with 1X TBST again, the protein signal was detected by chemiluminescence, using the Super Signal substrate (Pierce, Number: 34087).

### 2.8. Immunoprecipitation (IP)

After the indicated treatment, 293T cells were lysed with E1A lysis buffer (250 mM NaCl, 50 mM HEPES (pH 7.5), 0.1% NP-40, 5 mM EDTA, protease inhibitor cocktail (Roche)). Cell lysates were precleared with normal rabbit IgG (sc-2027, Santa Cruz Biotechnology, Dallas, Texas, United States) and protein A-agarose. Specific antibodies were added to the lysates (1 µg primary antibody/1 mg protein extract) and immunoprecipitated overnight at 4 °C. After washing the beads 4 times with E1A lysis buffer, the immunoprecipitated complexes were analyzed by immunoblotting.

### 2.9. Statistical Analyses

All values are presented as means ± standard error of the means of the replicate samples (n = 3). These experiments were repeated a minimum of three times. Differences between the two groups were assessed using the unpaired two-tailed Student's *t*-test. For testing the significance of pairwise group comparisons, Tukey's test was used. For all comparisons, *p*-values of < 0.05 were considered statistically significant. SPSS version 13.0 (SPSS Inc., Chicago, IL, USA) was used for all calculations.

## 3. Results

### 3.1. The Interaction between MRE11 and STAT3

To determine the role of MRE11 in the IL-6/STAT3 pathway, we investigated the effect of MRE11 on STAT3 Tyr705 phosphorylation (pSTAT3) through Western blotting. IL-6 induced STAT3 phosphorylation in A549 shLeu cells, but not in A549 shMRE11 cells (Figure 1A). Moreover, MRE11 interacted with STAT3 under IL-6 treatment (Figure 1B).

### 3.2. The Effect of MRE11 on Activation of STAT3 under Treatment with IL-6

Next, we observed the effect of MRE11 on the mRNA expression of these downstream genes. After the cells were treated with the indicated IL-6 concentration, total mRNA was extracted from the A549 shLeu and A549 shMRE11 cells. Our data demonstrated that IL-6 can induce the mRNA expression of STAT3 downstream genes, including *PLSCR1* (Figure 2A), *ID1* (Figure 2B), and *XBP1* (Figure 2C) in the A549 shLeu cells, but not in the A549 shMRE11 cells (Figure 2A–C).

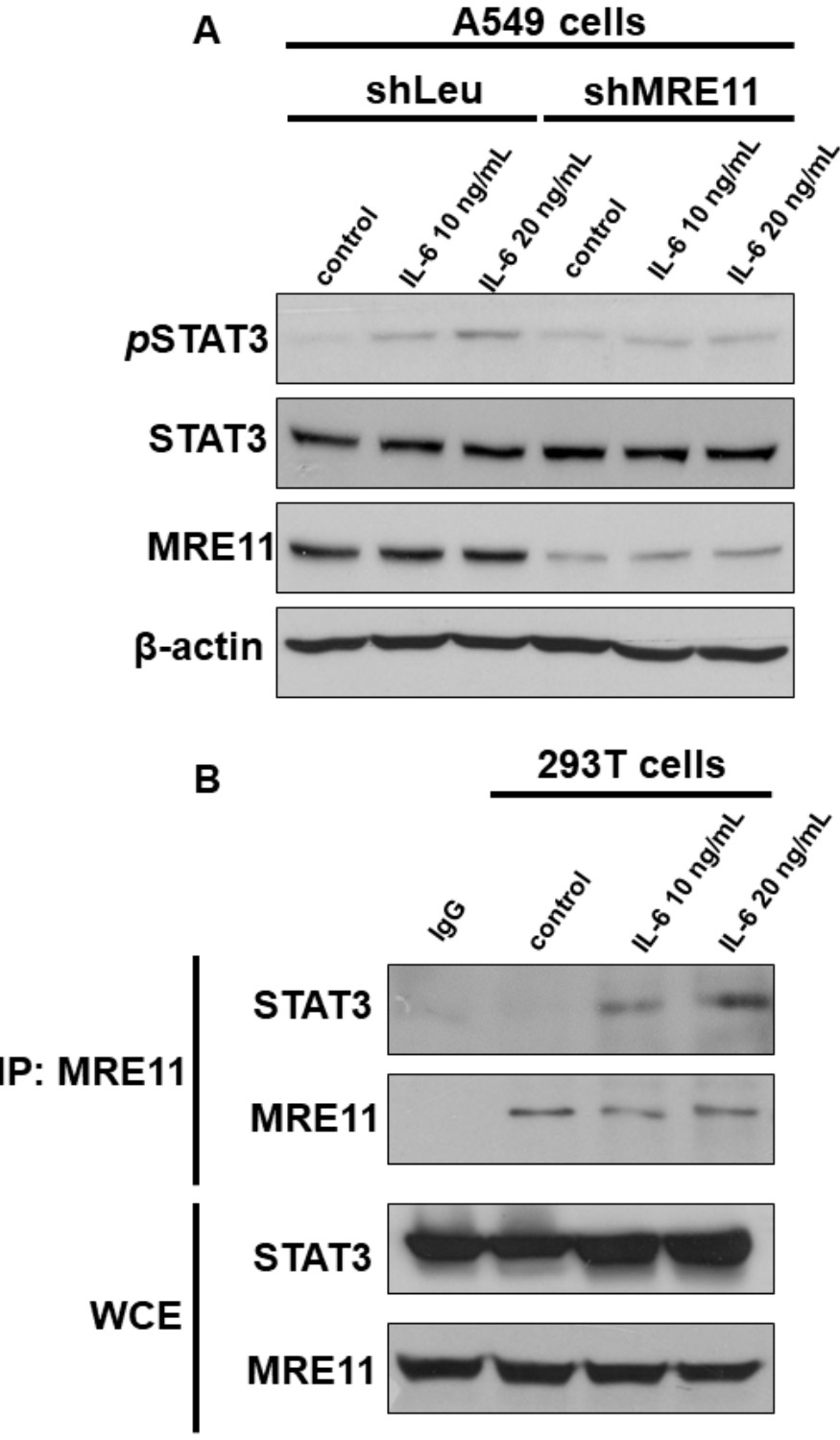

**Figure 1.** The effect of MRE11 on the protein expression of *p*-STAT3 and interaction with STAT3. (**A**) Total cell extracts of A549 shLeu and A549 shMRE11 cells were harvested from cells treated with IL-6 for 24 h. A549 cells silenced with MRE11 were treated with indicated concentrations of IL-6 for 24 h. The protein was immunoblotted with polyclonal antibodies specific for *p*-STAT3, STAT3, or MRE11. β-actin was used as an internal loading control. (**B**) After treatment with indicated concentrations of IL-6 for 24 h, 293T cells were harvested for immunoprecipitation with MRE11 antibody, followed by Western blotting analysis.

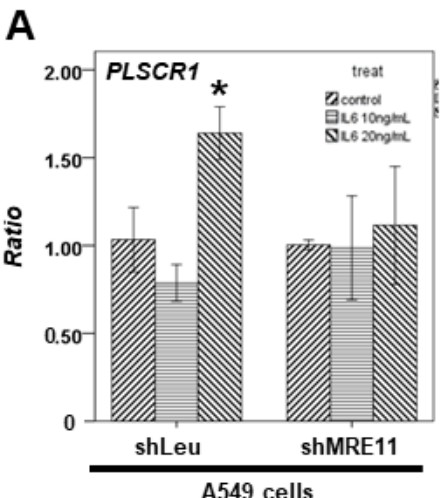

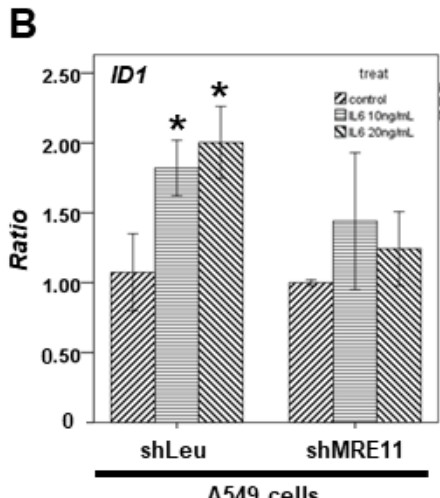

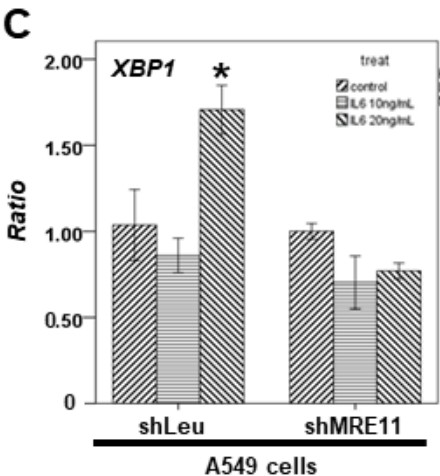

**Figure 2.** The effect of MRE11 on STAT3′s downstream genes treated with IL-6. Total mRNA was extracted from the A549 shLeu and A549 shMRE11 cells after treatment without or with IL-6 for 24 h. The coding regions of human *PLSCR1* (**A**), *ID1* (**B**), and *XBP1* (**C**) were used as probes for real-time polymerase chain reaction analysis. (Error bars = mean ± S.E.M. Asterisks (*) mark samples significantly different from the DMSO group with $p < 0.05$.).

### 3.3. The Effect of MRE11 on Cytokine Secretion from Lung Cancer Cells

Tumor cells secrete some molecules, such as chemokines, during invasion and migration. Next, we investigated the effect of MRE11 on cytokine secretion from lung cancer cells. The secreted cytokines from the A549 shLeu and A549 shMRE11 cells were screened using the human cytokine array. The results revealed that MRE11 deficiency can decrease the expression of the cytokines *CCL2*, CCL5, interleukin-1 receptor antagonist, G-CSF, IL-6, and intercellular adhesion molecule 1 (Figure 3A). A previous study showed that hypoxia can induce IL-6 production and subsequent secretion of *CCL2* [33,34]. Next, we investigated the effect of MRE11 on the IL-6/*CCL2* pathway under hypoxia. In a hypoxia environment for 24 h, we also discovered that secretion of IL-6 of A549 shMRE11 cells can be induced. However, the secretion of IL-8 and *CCL2* was inhibited in A549 shMRE11 cells (Figure 3B–D).

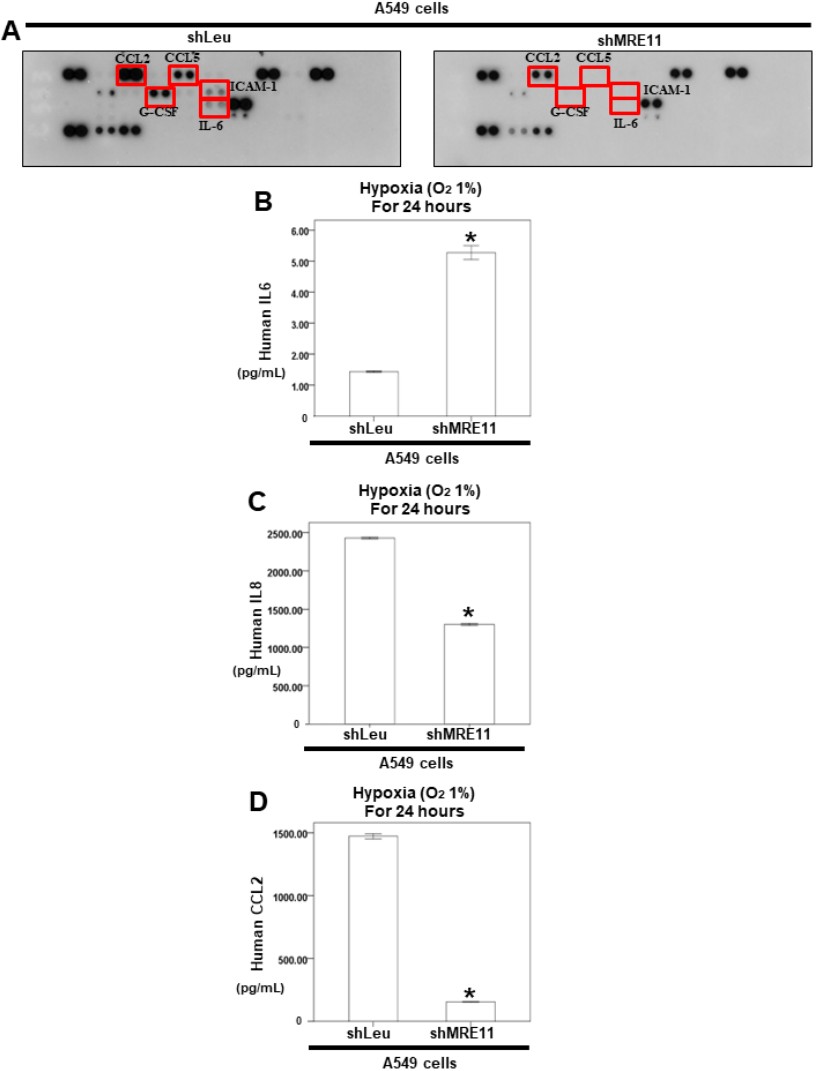

**Figure 3.** The protein secretion from A549 shLeu and A549 shMRE11 cells. (**A**) For cytokine array, the cultured medium of A549 shLeu and A549 shMRE11 cells were collected and analyzed by cytokine microarray. Array images were captured following a 5-min exposure to X-ray film. (**B–D**) For ELISA, the medium was collected from A549 shLeu and A549 shMRE11 cells after treatment with hypoxia for 24 h. The secretion of human IL-6, IL-8, or CCL2 was measured by ELISA kits. (Error bars = mean $\pm$ S.E.M. Asterisks (*) mark samples significantly different from the DMSO group with $p < 0.05$.).

### 3.4. The Effect of MRE11 on STAT3′s Downstream Genes from Lung Cancer Cells under Activation of Macrophages

Next, we treated the A549 shLeu or A549 shMRE11 cells with the conditional medium of THP-1 cells to simulate a co-cultured system. The qPCR results revealed that *VEGF*, *RUNX2,* and *LDHA* mRNA expression could be activated in conditional medium-treated A549 shLeu cells, but not in the treated A549 shMRE11 cells (Figure 4A–C).

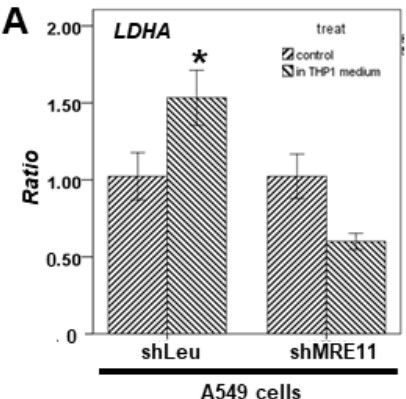

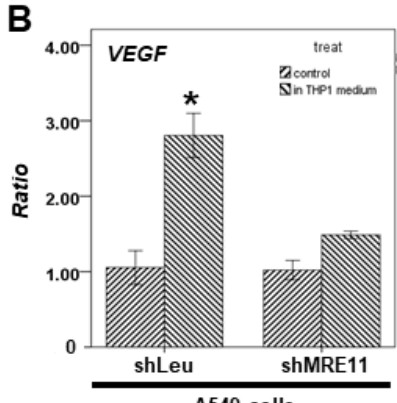

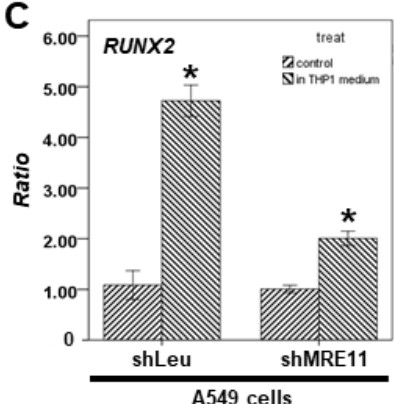

**Figure 4.** The effect of MRE11 on STAT3′s downstream genes treated with a conditional medium of macrophages. (**A–C**) Total mRNA was extracted from A549 shLeu and A549 shMRE11 cells after treatment with or without the conditional medium of THP-1 for 24 h. The coding regions of human *LDHA, VEGF,* and *RUNX2* were used as probes for real-time polymerase chain reaction analysis. (Error bars = mean ± S.E.M. Asterisks (*) mark samples significantly different from the DMSO group with $p < 0.05$.).

### 3.5. Effects of MRE11A on RAW 264.7 Cell Recruitment In Vitro

Our results showed that *CCL2* mRNA expression could be activated in a conditional medium of THP-1 cells-treated A549 shLeu cells, but not in the treated A549 shMRE11 cells (Figure 5A). Moreover, *CCL2* secretion was lower in the A549 shMRE11 cells than in the A549 shLeu cells after treatment with the conditional medium of THP-1 cells or RAW264.7 cells according to the ELISA results (Figure 5B,C).

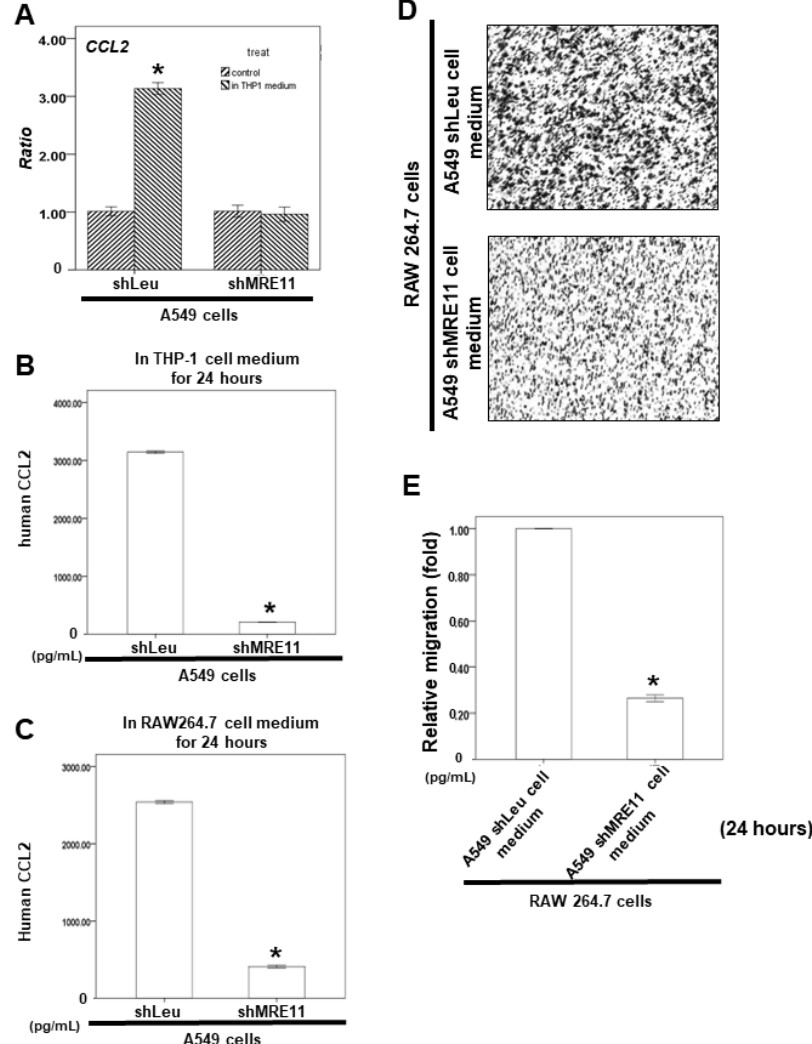

**Figure 5.** The effect of MRE11 on RAW 264.7 cell recruitment in an in vitro model. (**A**) Total mRNA was extracted from A549 shLeu and A549 shMRE11 cells after treatment with or without the conditional medium of THP-1 for 24 h. The coding regions of human *CCL2* were used as probes for real-time polymerase chain reaction analysis. (**B,C**) For ELISA, the medium was collected from A549 shLeu and A549 shMRE11 cells after treatment with or without the conditional medium of THP-1 or RAW264.7 cells for 24 h. The secretion of human CCL2 was measured by ELISA kits. (**D,E**) For the macrophages' recruitment ability of human lung cancer cells, the conditioned medium from A549 shLeu or A549 shMRE11 cells was collected and placed in the lower chamber. Then, RAW 264.7 cells were then placed in the upper chamber for the migration assay. (**D**) After incubation for 24 h, the photographs (×100) were taken, and the migratory cells were measured using AlphaEase®FC StandAlone Software. (**E**) The quantification of the migratory RAW 264.7 cell numbers in each group was normalized to the control. The results were from three independent experiments. (Error bar = mean ± S.E.M. Asterisks (*) mark samples significantly different from the blank group with $p < 0.05$).

We also investigated the effect of MRE11 on macrophage recruitment by lung cancer cells. The cancer cells were cultured for 24 h. The conditioned media were collected and placed in the lower chambers, and RAW 264.7 cells were placed in the upper chambers of transwell plates in a serum-free medium for a migration assay. These results indicated that MRE11 deficiency in lung cancer cells can decrease their ability to recruit RAW 264.7 cells (Figure 5D,E).

## 4. Discussion

In a previous study, mice administered anti-IL6 antibodies exhibited increased radiation-induced mortality [35]. Moreover, IL-6 silencing in human lung cancer cell lines resulted in higher DSBs after irradiation [36,37]. Further, a previous study showed that MRE11 knockdown suppressed IL-6 expression in mouse embryonic fibroblast cells [38]. In our results, MRE11-deficient A549 cells showed decreased IL-6 expression (Figure 3). In addition, MRE11 can interact with STAT3 under IL-6 treatment and can regulate STAT3 Tyr705 phosphorylation (Figure 1B). IL-6 cannot induce STAT3 activation without MRE11 (Figure 2). The MRE11 signaling pathway can also regulate *CCL2* secretion (Figure 4). These data indicate that MRE11 plays a critical role in IL-6 pathways. However, the effect of other cytokines on MRE11 must be investigated.

The IL-6/STAT3 pathway is hyperactivated in patients with many cancer types, including lung cancer [39]. In the tumor environment, IL-6 can activate STAT3 signaling in both cancer cells and tumor-infiltrating immune cells, including macrophages, and can promote the proliferation and metastasis of cancer cells [39]. A previous study showed that DSBs can activate the STAT3 signaling pathway [40]. Moreover, breast cancer cell lines with MRE11 overexpression can induce STAT3 phosphorylation at tyrosine-705 and serine-727 residues and promote cancer cell proliferation and migration [41]. In our study, IL-6 could not induce STAT3 Tyr705 phosphorylation in the A549 shMRE11 cells (Figure 1A). Moreover, MRE11 could interact with STAT3 in IL-6-treated cells (Figure 1B). On the other hand, a previous study showed that MRE11 interacted with dsDNA in the cytoplasm and not in the nucleus [38]. The location of the MRE11–STAT3 interaction must be investigated for a deeper understanding of the mechanism.

*CCL2*, an important cytokine for macrophage chemotaxis and activation [42], is also produced by many types of tumor cells [43]. DSBs can result in higher *CCL2* expression in cancer cells for macrophage activation and recruitment [44]. Moreover, lung cancer is a genomically unstable cancer with a high mutation rate [45]. Lung cancer patients with high *CCL2* expression have a poor prognosis [46]. *CCL2* signaling is the important pathway through which macrophages can activate the growth and metastasis of lung cancer cells by triggering bidirectional cross-talk between macrophages and cancer cells [15]. Our results showed that *CCL2* mRNA expression and *CCL2* secretion could not be activated in MRE11-knockdown lung cancer cells after macrophage activation (Figure 4). MRE11-knockdown A549 cells also exhibited a decreased ability to recruit macrophages (Figure 5). The results suggest that MRE11 could regulate the microenvironment of lung cancer through the *CCL2* pathway. On the other hand, the MRN complex plays a pivotal role in DSBs [1,2]. However, the effect of NBS1 and RAD50 on *CCL2* regulation remains unclear. The mechanism should be investigated further.

## 5. Conclusions

Together, our results suggest that IL-6 can induce the MRE11–STAT3 interaction and activate STAT3. Then, the activated STAT3 signaling pathway can induce *CCL2* secretion for macrophage recruitment. Furthermore, MRE11 is a potential target for lung cancer therapy.

**Author Contributions:** C.-Y.W. conceived the idea, designed the experiments, and wrote the main manuscript. L.-H.S., Y.-C.C. and H.-T.L. performed the experiments and prepared Figures 1–5. Y.-H.W. (Yu-Heng Wu) analyzed the data. Y.-H.W. (Yu-Huei Wu) revised the manuscript. All authors have read and agreed to the published version of the manuscript.

**Funding:** Financial support was obtained in the form of grants CMRPG6H0161, CMRPG6H0162, and CMRPG6H0163 from Chang Gung Memorial Hospital awarded to Ching Yuan Wu. The funding bodies allowed the design of the study, the collection, analysis, and interpretation of data, and the writing of the manuscript.

**Institutional Review Board Statement:** Not applicable.

**Informed Consent Statement:** Not applicable.

**Data Availability Statement:** All data generated or analyzed during this study are indicated in this article (with no patient data). The datasets generated during and/or analyzed during the present study are available from the corresponding author upon reasonable request.

**Acknowledgments:** The authors acknowledge the Health Information and Epidemiology Laboratory at the Chiayi Chang Gung Memorial Hospital for the comments and assistance in data analysis.

**Conflicts of Interest:** The authors declare no conflict of interest.

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
