# Peer review of "The Role of MRE11 in the IL-6/STAT3 Pathway of Lung Cancer Cells"

_cimb, doi:10.3390/cimb44120418_

Round 1

Reviewer 1 Report

This manuscript clearly describes a pivotal role of MRE11 in the IL-6/STAT3 pathway of lung cancer cell lines. These findings are insightful and would be of interest to readers of this journal. However, I consider that this manuscript has some problematic points, as follows.

Major points.

This manuscript seems to be lengthy, and should be shortened. Particularly in “Results”, needless sentences are scattered. For example, the authors described as “STAT3 is a critical protein in tumor metastasis, including in metastasis of lung cancer [7,8]. Persistently activated or tyrosine-phosphorylated STAT 3 (pSTAT3) is found in 50% of lung adenocarcinoma [16].” in “Results” (page 4, line 157). However, such findings are unnecessary in “Results”, and should be stated in “Introduction”, and/or “Materials and Methods”. Similarly, the authors described as “Professor Christine Watson reported that inhibitor of DNA binding 1….” (page 4, lines 168-172). Such findings should be described in “Introduction” or “Materials and Methods”. “Macrophages can promote lung tumor invasion …” (page 5, lines 194-); “CCL2 signaling pathway is pivotal for ….” (page 5, lines 209-), “Macrophages can migrate and … “ (p0age 5, lines 216-), …… are also unnecessary in “Results”. Similar findings are recurrently described in “Discussion”. In addition, the authors described “These results suggest that ….” (page 4, lines 176-177; page 4, lines 189-190), … These descriptions should be described in “Discussion”.

Minor points:

1)      The title: …” IL-6/STAT 3 Pathway…”  (rather than “…Il-6/STST3…”)

2)      .. activations ... (rather than “.. ac-tivations .. page 1, line 13)

3)      Abrupt abbreviation: What is “ATM”? (page 1, line 30), What is “ATR kinase”? (page 1, line 32), “CCL2” (page 1, line 45; its full-name is described after this abbreviation),

Author Response

Reviewer #1: This manuscript clearly describes a pivotal role of MRE11 in the IL-6/STAT3 pathway of lung cancer cell lines. These findings are insightful and would be of interest to readers of this journal. However, I consider that this manuscript has some problematic points, as follows.

Major points.

This manuscript seems to be lengthy, and should be shortened. Particularly in “Results”, needless sentences are scattered. For example, the authors described as “STAT3 is a critical protein in tumor metastasis, including in metastasis of lung cancer [7,8]. Persistently activated or tyrosine-phosphorylated STAT 3 (pSTAT3) is found in 50% of lung adenocarcinoma [16].” in “Results” (page 4, line 157). However, such findings are unnecessary in “Results”, and should be stated in “Introduction”, and/or “Materials and Methods”.

=>Response:

We thank the reviewer’s comment. We have rewritten “Introduction” and “Results” in the revised manuscript depending on the reviewer’s comments.

Similarly, the authors described as “Professor Christine Watson reported that inhibitor of DNA binding 1….” (page 4, lines 168-172). Such findings should be described in “Introduction” or “Materials and Methods”. “Macrophages can promote lung tumor invasion …” (page 5, lines 194-); “CCL2 signaling pathway is pivotal for ….” (page 5, lines 209-), “Macrophages can migrate and … “ (p0age 5, lines 216-), …… are also unnecessary in “Results”.

=>Response:

We thank the reviewer’s comment. We have rewritten “Introduction” and “Results” in the revised manuscript depending on the reviewer’s comments.

Similar findings are recurrently described in “Discussion”. In addition, the authors described “These results suggest that ….” (page 4, lines 176-177; page 4, lines 189-190), … These descriptions should be described in “Discussion”.

=>Response:

We thank the reviewer’s comment. We have rewritten “Discussion” and “Results” in the revised manuscript depending on the reviewer’s comments.

Minor points:

1) The title: …” IL-6/STAT 3 Pathway…”  (rather than “…Il-6/STST3…”)

=>Response:

We thank the reviewer’s comment. We have rewritten the title in the revised manuscript depending on the reviewer’s comments.

2). activations ... (rather than “.. ac-tivations .. page 1, line 13)

=>Response:

We thank the reviewer’s comment. We have rewritten the line in the revised manuscript depending on the reviewer’s comments.

3) Abrupt abbreviation: What is “ATM”? (page 1, line 30), What is “ATR kinase”? (page 1, line 32), “CCL2” (page 1, line 45; its full-name is described after this abbreviation),

=>Response:

We thank the reviewer’s comment. We have rewritten the lines in the revised manuscript depending on the reviewer’s comments.

Reviewer 2 Report

The manuscript uncovered a linked between MRE11 and IL-6/STAT3 14 pathway.

MRE11 is pivotal protein for ATM activation during double strand DNA break. ATM kinase activations may act as lung cancer biomarkers. IL-6/STAT3 pathway play important role in tumor metastasis, including lung cancer. 

Together, the results suggest that IL-6 could induce the MRE11–STAT3 interaction and activate STAT3. This is significant as  the activated STAT3 signaling pathway could induce CCL2 secretion for macrophage recruitment as shown using RAW cells .

The data are clean and convincing and overall fits the requirements for publication in this journal .  The only comment that could be made in  re-enforcing the message is that key  data ( interactions or KD -CCL2 CCL5 effect   )  can also be shown using additional lung cell lines . 

Author Response

Reviewer #2: The manuscript uncovered a linked between MRE11 and IL-6/STAT3 14 pathway.

MRE11 is pivotal protein for ATM activation during double strand DNA break. ATM kinase activations may act as lung cancer biomarkers. IL-6/STAT3 pathway play important role in tumor metastasis, including lung cancer.

Together, the results suggest that IL-6 could induce the MRE11–STAT3 interaction and activate STAT3. This is significant as  the activated STAT3 signaling pathway could induce CCL2 secretion for macrophage recruitment as shown using RAW cells .

The data are clean and convincing and overall fits the requirements for publication in this journal . 

The only comment that could be made in  re-enforcing the message is that key data ( interactions or KD -CCL2 CCL5 effect )  can also be shown using additional lung cell lines .

=>Response:

We thank the reviewer’s comment. Because the revised duration is 7 days, there is no enough time to perform these experiments. However, the proposal for the deeper mechanism between MRN complex and immune response is going on in our lab. We will strengthen key data depending on reviewer’s comments.

Round 2

Reviewer 1 Report

 No spaces between words and parentheses, such as “phages[5]” (page1, line 39); “.. cancer[22,23]” (page 2, line 64)….

Author Response

Reviewer #1: Comments and Suggestions for Authors

 No spaces between words and parentheses, such as “phages[5]” (page1, line 39); “.. cancer[22,23]” (page 2, line 64)….

=>Response:

We thank the reviewer’s comment. We have rewritten the revised manuscript depending on the reviewer’s comments.
